# Challenges for the Newborn Immune Response to Respiratory Virus Infection and Vaccination

**DOI:** 10.3390/vaccines8040558

**Published:** 2020-09-24

**Authors:** Kali F. Crofts, Martha A. Alexander-Miller

**Affiliations:** Department of Microbiology and Immunology, Wake Forest School of Medicine, Winston-Salem, NC 27101, USA; kcrofts@wakehealth.edu

**Keywords:** newborn, respiratory viruses, respiratory infection, vaccines, innate immunity, adaptive immunity

## Abstract

The initial months of life reflect an extremely challenging time for newborns as a naïve immune system is bombarded with a large array of pathogens, commensals, and other foreign entities. In many instances, the immune response of young infants is dampened or altered, resulting in increased susceptibility and disease following infection. This is the result of both qualitative and quantitative changes in the response of multiple cell types across the immune system. Here we provide a review of the challenges associated with the newborn response to respiratory viral pathogens as well as the hurdles and advances for vaccine-mediated protection.

## 1. Introduction

Respiratory viral infections place a significant burden on human health globally. There are an estimated 11.9 million cases of acute lower respiratory tract infections (ALRI) in young children every year, with over half of these cases occurring in infants [1]. Moreover, there is a three-fold increase in the fatality rate following infection in infants under 12 months old compared to children aged 12–59 months old [1]. The viruses that account for the majority of hospitalizations in infants suffering from lower respiratory tract infections include influenza virus, respiratory syncytial virus (RSV), rhinoviruses (RVs), and human parainfluenza viruses (HPIVs) [2]. Newborns often experience the most severe disease following infection with these pathogens. As infants age and their immune system matures, the disease state experienced typically lessens.

The increased susceptibility to severe disease in newborns following respiratory virus infection is a result of their naïve immune status together with the altered responsiveness of the immune system. The altered response in newborns is in part due to the need to create a permissive environment for establishment of the microbiome. Colonization by the array of microbes that make up the microbiome is an important regulator of immune system development [3]. The inherent bias of the neonatal immune system towards an anti-inflammatory T cell response allows for colonization of the microbiome without generating potent pro-inflammatory signals that can lead to pathogenic, tissue-damaging responses [3]. Although the newborn benefits from an immune state that allows establishment of the microbiome, the trade-off is an immune system that often contributes to increased susceptibility to severe disease following viral infection.

Currently, there are no vaccines available against the pathogens most important for newborn health (influenza virus, RSV, RVs, or HPIVs) for infants under 6 months of age. Given the toll of these viruses, there is a pressing need to develop safe and efficacious vaccines for this vulnerable population. The major challenge associated with neonatal vaccination is overcoming the immaturity of the immune system while avoiding potent pro-inflammatory signals that promote inflammation resulting in tissue damage, allergy, and/or autoimmunity [4,5]. Here, we will discuss the challenges associated with the neonatal immune system in response to respiratory viral infections and the advancements in the generation of effective vaccination strategies for this at-risk population.

## 2. The Neonatal Immune System

Alterations in the neonate immune system span the innate and adaptive arms, both of which are vital contributors to the clearance of viral infections. Defects associated with the innate immune system are varied and include reduced activity of innate pattern recognition receptors, decreased interferon (IFN) production from plasmacytoid dendritic cells (PDCs), poor conventional dendritic cell (cDC) maturation and cytokine production, reduced cellular migration, impaired phagocytosis, and impaired NK cell function. Adaptive defects include reduced T cell activation and effector function, altered T cell differentiation, impaired B cell differentiation and survival, and decreased antibody quantity and quality (Figure 1). Together, these properties of the newborn immune system result in compromised protection against viral infections and disease. Each is discussed in detail below.

### 2.1. Innate Immune System

The newborn innate immune system is the first line of defense against infection [6]. Among the cells involved in the early innate response are monocytes, macrophages, natural killer (NK) cells, DC, and neutrophils. These cells are important in pathogen clearance and are also essential in initiating an adaptive immune response, i.e. DC [7]. Activation of these cells often occurs by pattern recognition receptor (PRR) signaling, e.g. toll-like receptors (TLRs) and retinoic acid-inducible gene I (RIG-I), following binding of pathogen associated molecular patterns (PAMPs) present in the invading pathogen. Numerous reports show that both TLRs and RIG-I mediated responses are impaired in neonates (e.g. [8,9,10,11,12]). The defects in these innate receptors also affect generation of the adaptive immune response as they are important for maturation and activation of professional antigen presenting cells (APCs).

Monocytes and macrophages mediate leukocyte recruitment, phagocytose pathogens, and regulate inflammation. Interestingly, monocyte counts are higher in newborns than adults. Infants are reported to reach adult numbers by 3–5 months of age [13], although others suggest it is later [14]. Monocytes derived from newborns exhibit impairments in many functions including chemotaxis, phagocytosis, responsiveness to TLR engagement, and adhesion molecule expression (LFA-1 and LFA-2) [15,16,17]. Full competency in these functions occurs between 6 and 12 months of age [14].

Human DCs are divided into two developmentally-distinct lineages, cDC and PDCs. cDCs respond to pathogen signals by undergoing a process called maturation, which makes them competent for T cell activation. cDCs from newborns are challenged in this process, exhibiting a decreased capacity to respond to PAMPs [9,11,18,19,20,21,22], reduced expression of costimulatory molecules (CD40, CD80 and CD86) [9,11,23,24,25,26], and diminished IL-12p70 production [11,19,20]. The altered function of cDC is present through at least 3 months of age and may not be adult-like until 12 months of age [14]. In addition to the impaired functional capabilities, studies support a decreased number of cDC available to activate T cells. For example, there is a reduction in the number of cDCs in human cord blood compared to adult blood [23,27]. Analysis of the spleen and lungs of infant mice also demonstrate a decrease in this important cell type [28,29].

In contrast to cDCs, PDCs are less involved in activation of T cells, but instead are important producers of type I interferon (IFN), an important contributor to the effective control of respiratory viral infections [30]. As with cDCs, PDCs in newborns are altered. Newborns make a significantly lower amount of type I IFN following infection as a result of decreased IFN production on a per-cell basis and a reduced number of PDCs [8,9,31,32].

NK cells are innate lymphocytes that contribute to the clearance of viruses as well as other pathogens [33,34]. NK cells lyse virus-infected cells lacking major histocompatibility complex I (MHC I) molecules [34] as well as antibody coated cells via a process called antibody-dependent cellular cytotoxicity (ADCC). NK cells are also important for production of an array of immunomodulatory cytokines [35]. Interestingly, NK cells are higher in number compared to adults immediately following birth [14]. They decline rapidly over the first postnatal days (2–3-fold) followed by a more modest progressive reduction until they reach adult levels by the time children reach 5 years of age [36]. However, although their numbers are not reduced, these cells show significant alterations in function, i.e. reduced cytolytic capacity [37]. This may be, at least in part, the result of lower expression of the cytolytic mediator granzyme B in newborns compared to adult-derived NK cells [38]. Newborn NK cell function may also be impaired as a result of the reduced expression of ICAM-I, a molecule important for cell adhesion [39,40]. Cytolytic activity in NK cells increases steadily after birth, approximating adult levels by 5 months of age [41].

Neutrophils are one of the first cells to enter the site of infection. They promote elimination of microbes through phagocytosis and the release of neutrophil extracellular traps (NET). Neutrophils seem to be present at adult-like numbers in newborns [42,43,44]. Studies diverge on the extent to which neutrophil function is altered in these individuals [43,44]. The impaired ability of newborn neutrophils to take up non-opsonized pathogens has been reported [45]. It is likely that the pathogen studied and whether there is antibody or complement present to facilitate uptake are responsible for the differences observed. Newborn neutrophils also appear to exhibit impaired transmigration through vascular endothelium which is associated with reduced adhesion molecule levels [42,43]. Reports differ with regard to the age at which neutrophils reach adult levels of migration, reporting ranges from 4 weeks to 12 months [14]. Phagocytic activity has been suggested to require around 12 months to fully reach adult levels [14].

γδ T cells can play an important role in viral clearance through the release of cytokines as well as cytolytic factors [46]. These cells recognize a broad range of antigens associated with infection, e.g. stress induced molecules [46]. An elegant study by Gibbons et al. compared γδ T cells from term infants and adults [47]. They found that infant γδ T cells were highly functional, producing similar amounts of IFNγ following activation as adult γδ T cells [47]. These findings suggest γδ T cells may be wired to compensate for a weak T helper 1 (TH1) response that is often present in newborns.

In summary, the studies of the neonate innate immune system have identified the presence of multiple alterations that can contribute to the decreased ability of this age group to clear respiratory viral infections.

### 2.2. Adaptive Immune System

The adaptive immune system is key to generating an antigen-specific response that effectively clears infection. It is also responsible for the establishment of immunological memory, which provides protection following re-exposure to the pathogen. Generation of a T cell response is dependent on antigen presentation by cDCs that have undergone maturation. The reduced ability of cDCs from newborns to upregulate expression of costimulatory molecules [9,11,23,24,25,26] and the diminished IL-12p70 production [11,19,20] poses a significant hurdle for T cell activation. The reduction in IL-12 together with the increase in IL-4 in newborns is responsible for directing activated CD4^+^ cells along the TH2 pathway [4,48,49]. The resulting impaired TH1 response and enhanced TH2 response is a contributing factor to the increased susceptibility to viral infections in newborns [18,50]. Although an initial antigenic stimulation can induce both TH2 and TH1 responses in neonatal mice, upon re-exposure to antigen, TH1 cells undergo apoptosis [51]. The loss of TH1 cells is the result of the upregulation of IL-13Rα1, which disables the balance of pro- and anti-apoptotic factors [51]. Blockade of IL-13Rα1 and IL-4Rα restores the TH1 response [51]. Decreased IL-12p70 production by APCs also negatively impacts CD8^+^ cytotoxic T lymphocyte differentiation and acquisition of effector function [52]. The lower costimulatory signals present on APCs leads to diminished CD28 engagement [29], which also contributes to impaired T cell activation in neonates.

Compounding the effects of alterations in cDCs, neonatal T cells exhibit inherent defects in activation and differentiation [53,54,55,56,57,58,59]. Impairments have been reported across multiple T cell subsets including CD4^+^ T cells, CD8^+^ T cells, and T follicular helper cells [20,29,51,52,60,61]. T cells recognize the presence of pathogens through the binding of their T cell receptor (TCR) to pathogen-derived peptides presented in complex with major histocompatibility complex (MHC) molecules, termed HLA in humans. Diminished neonatal T cell activation is associated with decreased levels of the molecules important in initiating signaling through the TCR, e.g. lck and ZAP-70 [59]. Other studies have reported a reduction in AP-1 mediated transcription in neonatal versus adult T cells [62]. AP-1 is involved in the expression of multiple genes in T cells, including IL-2 which drives proliferation [63].

Tfh cells are necessary for the generation of high level, high affinity antibody secreting cells. In keeping with other CD4^+^ T cell types, Tfh responses are significantly impaired in newborns [60,61]. Tfh cells promote B cell responses through expression of CD40L as well as the production of cytokines (IL-21 together with IL-4, IFNγ, or IL-17) [64,65]. The result of these signals is the generation of germinal centers (GC), class switching, and affinity maturation [61]. In a study comparing Tfh cells and GC B cell responses in 1 week old mice and adult mice, vaccination with aluminum hydroxide adjuvanted tetanus toxoid (TT) resulted in a decrease in the expansion of Tfh cells, limited GC B cells, and decreased TT-specific IgG in newborns compared to adults [60]. Debock et al. vaccinated neonatal (7 day old) and adult (10 week old) mice with ovalbumin-aluminum hydroxide and boosted after 3 weeks. Here again, vaccinated neonatal mice had impaired Tfh responses that led to decreases in antibody production, poor affinity maturation, and delayed GC reactions [61].

Finally, neonates have heightened T regulatory cell (Treg) responses. Tregs are more highly represented in the circulation of human infants compared to adults [37,66,67,68,69,70,71]. Our studies have shown this to be true in nonhuman primate newborns as well [72,73]. The higher representation of these cells has significant implications for immune regulation. Tregs are important players in the suppression of anti-maternal immune responses [74] and are thought to contribute to the dampening of inflammatory responses generated by the establishing microbiome [75]. While these effects are undoubtedly of benefit, the increase in Tregs can limit virus-specific T cell generation in response to infection [76,77,78]. Interestingly, the increased abundance of Tregs is at least in part the result of increased propensity of newborn T cells to differentiate into FoxP3^+^CD25^+^ Tregs [71,74,78,79,80].

There is also a variety of impairments in the humoral arm of the newborn adaptive response ranging from B cell differentiation and survival to a significant decrease in antibody. In humans, it is apparent that IgG production is weak during the first year of life and although IgM production is greater than IgG during this period, it is also suboptimal [81,82,83]. This weak antibody response is likely a result of both the impaired Tfh cell responses described above and autonomous defects in B cells. Analysis of neonate B cells revealed reduced expression of BCMA and BAFF-R, two important signals in driving B cell differentiation and survival. Specifically, engagement of these two molecules promotes survival through the upregulation of anti-apoptotic bcl-2 family members and downregulation of the pro-apoptotic factors bim and bad [84]. There are also changes in accessory cells that support antibody secreting cells. Expression of APRIL on bone marrow stromal cells is important for the survival of plasmablasts and differentiation into long-lived plasma cells [85,86]. The decreased expression of APRIL in neonate stromal cells is therefore a significant barrier to long-lived antibody responses in newborns [85,87].

## 3. The Newborn Response to Respiratory Virus Infection

Respiratory viral infections in infants are among the leading causes of morbidity and mortality in this age group globally [88]. Major respiratory viruses that result in increased hospitalization in infants include influenza virus, RSV, RVs, and HPIVs [2]. The impairments observed in the innate and adaptive arms of the newborn immune system described above contribute to the increased susceptibility of these individuals to severe disease following viral infection. Below we review what is known about the nature of the immune response generated in newborns following infection with these clinically relevant respiratory viral pathogens.

### 3.1. Influenza Virus

Influenza virus infection results in 374,000 hospitalizations globally in infants under the age of 1, with approximately 72% of these hospitalizations occurring in infants under the age of 6 months [89]. Disease states associated with influenza virus infection in infants and children include otitis media, pneumonia, myositis, and croup. The last is restricted primarily to individuals less than 1 year of age and can be life threatening. The risk of lower respiratory disease is significantly increased in children less than 2 years of age [90,91,92,93,94]. In addition, bacterial pneumonia, which contributes to the increased lower respiratory disease in this age group, is a common complication of influenza virus infection [95]. Bacterial coinfection has been shown to be a significant predictor of severe disease requiring admission to the pediatric intensive care unit [96]. The increased susceptibility of newborns to disease following influenza virus infection has also been demonstrated in animal models, e.g. mice and nonhuman primates (NHP) [72,97,98]. Thus, from an experimental standpoint, animal models have proven a tractable approach for probing the mechanistic basis of the influenza virus disease in newborns.

As influenza virus is a relatively localized infection, the innate cells present in the airways are important players in the initial defense against virus. Epithelial cells, alveolar macrophages (AM), neutrophils, and NK cells are all involved in the innate attack against influenza virus. Epithelial cells are the primary target of influenza virus infection in the respiratory track. These infected cells are important for the production of type I IFN as well as cytokines and chemokines (IL-6, TNF-α, IL-8, CXCL8, CXCL10, CCL2, and CCL5) that trigger the recruitment of other immune cells [99,100,101,102,103]. Infant epithelial cells have been shown to have functional deficits in response to viral infection [104]. A study by Clay et al. identified the differences between epithelial cells infected with H1N1 influenza virus in infants compared to adults. The data revealed an increase in viral replication coupled with a decrease in type I IFN production [104].

Analysis of umbilical cord NK cells exposed to influenza virus revealed a decrease in perforin expression as well as increased apoptosis compared to adult derived NK cells [105]. While not studied in an in vivo model, to our knowledge, the functional impairments in newborn NK cells make them likely contributors to the increase in influenza disease.

CD8^+^ T cells are essential in the clearance of influenza virus in adult mice [106,107]. Impaired CD8^+^ T cell responses contribute to the increased susceptibility to disease observed following influenza virus infection in neonates [108]. In a study by You et al., 7-day old mice were infected with a mouse-adapted influenza A virus strain (A/PR/8/34 (H1N1) (PR8)). The authors found a decrease in the number of IFNγ-producing CD8^+^ T cells in the lungs compared with infected 4-week old mice [109]. A decrease was also observed in the CD4 compartment, albeit less (2-fold) than that observed for CD8^+^ T cells. In addition to a reduction in virus-specific T cells, the authors found that these effector cells were reduced in their ability to clear virus as demonstrated by adoptive transfer of neonate vs. adult T cells [109]. The reduced generation of a T cell response may be in part the result of poor maturation of cDCs. Studies of umbilical cord cells revealed reduced expression of costimulatory markers (CD40, CD80 and CD86) and HLA-DR as well as limited IFN-α production in influenza virus-infected DC compared to adult peripheral blood cDCs [110]. Stimulation with infected newborn DC resulted in decreased T cell proliferation compared to that observed with adult cDCs [110].

Since influenza virus infection is localized, it is important that T cells can migrate effectively to the lungs and airways to clear virus. Neonatal mice infected with influenza virus showed a 4-day delay in T cell migration to the lungs compared to adult mice [98]. This was coupled with reduced IFNγ production, which has been shown to promote this activity. Neonatal T cells that did reach the lungs were unable to efficiently migrate into the airways. The authors proposed this was the result of an observed reduction in CXCL9 and CCL2. These studies highlight the challenges newborns face in generating potent effector anti-influenza T cell responses and trafficking of these cells to the site of infection.

Given the higher representation of Tregs in newborns versus adults [37,66,67,68,69,70,71], investigators have evaluated the role of these cells in the context of influenza virus infection. Garvy and colleagues showed depletion of Tregs by treatment with anti-CD25 antibody resulted in an increase in activated CD4^+^ cells, including TH2s [111]. Similar effects were observed in FoxP3 deficient mice that lack Tregs. Surprisingly, loss of Tregs was accompanied by poorer viral clearance, presumably as a result of the increased number of TH2 cells present. These data show the regulatory effects mediated by Tregs in the newborn are complex and the effect of their depletion is dependent on the nature of the T cell response elicited.

In adults, the main antibody isotypes that are produced in response to an influenza virus infection are IgM, IgA, and IgG [112]. IgM is produced early during infection and is important in activating the classical complement pathway. IgA produced locally after infection is present at the mucosal surface of the respiratory tract and plays an important role in virus neutralization [113]. IgG contributes to viral clearance through both neutralizing and non-neutralizing activities [114,115] and is critical for long-lived protection [116]. Surprisingly, little work has been carried out on the antibody response to influenza virus infection in the neonatal mouse model. However, we have used the non-human primate (NHP) model to explore these questions given their similarity to humans in lung structure [117] and immune system development. In our initial studies, we showed neonatal NHP (6–10 days old) infected with influenza virus had increased lung pathology and higher viral load compared to adult NHP [72]. This was accompanied by decreased virus-specific IgG levels in the lung and bronchus-associated lymphoid tissue (BALT) at 14-days postinfection, although systemic IgG levels at this timepoint were unexpectedly similar to adults [72]. The formation of BALT following influenza virus infection serves as a local site for the generation of an influenza virus-specific immune response, which in mice has been shown to be an important contributor to viral clearance [118]. The data from the NHP newborn model suggest a decrease in local immune responses in the respiratory tract of neonates contributes to an increase in viral disease and reduced viral clearance.

Specificity is also important in determining the efficacy of an antibody response to influenza virus infection, as there are epitope-dependent differences in clearance mechanisms and cross-strain reactivity [119,120,121,122,123,124]. Studies from Yewdell and colleagues in the mouse model showed antibodies against influenza virus target five distinct antigenic sites in the globular domain of the hemagglutinin (HA) molecule [125]. The relative representation of antibodies to the individual sites on HA within the HA-specific antibody pool generated was found to evolve over time. Further, the nature of the antigen exposure, i.e. infection versus vaccination, resulted in differences in the relative representation of antibodies that bind to the five antigenic sites. Our studies in the NHP model revealed altered epitope dominance patterns in the early IgM anti-HA response in newborns compared to adults [126]. However, over time these patterns became similar. Importantly, the IgG patterns did not differ between neonates and adults [126]. Our studies also investigated the ability of newborns to generate HA stem-specific antibodies. This region is relatively conserved in the HA molecule and thus these antibodies are capable of protecting against multiple virus strains. We found that these antibodies were readily generated in newborn NHP [126,127], suggesting the repertoire of the newborn is capable of mounting this response.

From the above data it is clear that multiple impairments associated with the neonatal immune system contribute to the increased vulnerability of this population to severe disease following influenza virus infection. As such, it will be essential that we understand which of these is most important for determining efficacy of the response and importantly, is a target for safe and effective modulation.

### 3.2. Respiratory Syncytial Virus

Respiratory syncytial virus (RSV) is a leading cause of respiratory tract infections and hospitalizations in young children, with infants under 3 months of age being the most seriously affected [128]. The toll of RSV infection on the health of children is high, with 11,800 deaths in individuals <5 years of age [129]. Infants infected with RSV are at high risk of viral pneumonia and bronchiolitis [130,131]. Those experiencing lower respiratory tract disease present with wheezing, crackles, tachypnea, hypoxia, nasal flaring, and other signs of respiratory distress [130]. Finally, infection with RSV during infancy has been identified as a risk factor for the development of childhood asthma [132].

Multiple innate populations are involved in early control of RSV infection including neutrophils, PDCs, AM, and NK cells. However, the innate system can be a dual-edged sword in the case of RSV in the newborn as it can also contribute to increased pathology. A study by Mejias et al. analyzed whole blood transcriptional profiles to characterize the immune response from infants hospitalized due to RSV infections [133]. This analysis revealed a dysregulated immune profile induced by RSV infection, showing overexpression of genes associated with neutrophil activation, type I IFN and systemic inflammatory markers when compared to healthy age-matched infants [133]. Further, infected infants exhibited reduced expression of genes regulating T and B cell activation [133].

RSV infection of airway epithelial cells results in neutrophil activation and recruitment to the site of infection. Recruited neutrophils release chemokines, cytokines, superoxide, and enzymes that can result in cell damage [134]. Severe bronchiolitis in infants is associated with this potent influx of activated neutrophils in the airways [135,136]. In infants suffering from RSV induced bronchiolitis, neutrophils account for greater than 80% of inflammatory cell infiltrate in the bronchoalveolar lavage (BAL) during the peak of symptoms [136].

PDC production of type I IFN during RSV infection is critical for viral clearance, NK cell expansion and activation, and limiting immune cell-mediated pathology [137,138]. Work by Cormier et al. found limited production of type I IFN and a reduction in PDC responses in RSV-infected newborn compared to adult mice [139]. The regulation of NK responses may play a role in enhanced disease in humans as a study by Welliver et al. analysing lung tissue sections from infants who had died from RSV infection found that severe bronchiolitis correlated with decreased NK cell infiltration [108].

Both CD4^+^ and CD8^+^ T cells contribute to effective RSV clearance [140,141,142]. The immunoregulatory processes at play in newborns have been shown to alter RSV-specific CD8^+^ T cells, both in number and epitope-specificity [29]. Altered dominance among responding T cells with regard to the epitope recognized has the potential to impact clearance, although this was not directly tested and as such, the implications of this change await further study. Mechanistically, the difference in the CD8^+^ T cell response was associated with alterations in both CD11b^+^ and CD103^+^ DCs, i.e. they are less effective in the uptake and processing of soluble antigen and express lower levels of costimulatory molecules [29].

The differentiation choice in CD4^+^ T cells responding to RSV infection has significant implications for clearance and disease. Studies assessing cytokines in the nasal and lung lavage of neonates suggest an altered T helper balance, i.e. decreased IFNγ vs. IL-4 [143,144,145,146]. TH2 cells are particularly deleterious in the context of RSV infection because of IL-4-mediated switching to IgE [147]. This antibody triggers mast cell activation and histamine release that promotes inflammation and bronchospasms [148]. In contrast, IFNγ downregulates IgE production [149], thereby limiting these effects. Increased differentiation into TH17 cells in infants is also associated with greater pathology following RSV infection [150]. The deleterious effects of IL-17 include neutrophilic lung inflammation, IL-13 production, mucus hypersecretion, and inhibition of CTL responses [151,152]. Loss of IL-17 reversed these effects and resulted in reduced pathology, demonstrating a causative role for this cytokine in disease.

A recent study in human newborns has shown a role for B regulatory cells (Bregs) in mediating disease following RSV infection [153]. The polyreactive nature of the BCR expressed by these cells allowed for binding to the RSV F protein [153]. The resulting activation of Bregs induced expression of CX3CR1 that bound the RSV G protein, promoting virus infection of the cell. The outcome of this was IL-10 production leading to a dampened TH1 response. Together, the studies described here lead to a model where alterations in both the innate and adaptive arms of the newborn immune response promote increased susceptibility to infection and disease severity in this age group following RSV infection.

### 3.3. Other Respiratory Viruses

While influenza virus and RSV represent major challenges for newborn health, two other respiratory viruses, RVs and HPIVs, merit discussion. With that said, much less is known about the immune response generated following infection with these viruses in newborns. Here we review our current understanding based on what has been reported.

RVs are the causative agent of the majority of colds and are implicated in the development of asthma and exacerbated wheezing in infants requiring hospitalization [154,155,156]. There are three distinct RV species- RV-A, RV-B and RV-C, with each having multiple subtypes. A study by Schneider et al. investigated the effects of RV1B infection in neonates and how this correlates with the development of asthma [157]. Seven-day-old BALB/c mice were infected with RV1B resulting in a rapid type I IFN response [157]. Twenty-eight days postinfection, the mice were assessed for asthma related symptoms. The authors found increases in CD4^+^ T cells producing IFNγ and NKT cells expressing IL-13 compared to controls, which correlated with increased airway hyper-responsiveness and mucus production. This was not observed in infected adult animals. When IL-13 was blocked in infants, hyper-responsiveness and mucus production were reduced [157]. These data suggest that the immune response that is generated as a result of RV infection in neonates can promote development of asthma-like symptoms.

HPIVs have been reported to account for 20–40% of paediatric hospitalizations of infants with lower respiratory tract illnesses [158]. There are four serotypes that cause disease in human infants, with HPIV3 accounting for 52% of the cases in the U.S. [159]. Newborns infected with HPIVs can develop a wide range of disease states including otitis media, croup, bronchiolitis, and pneumonia [158]. A correlate of protection against HPIV disease in adults is the presence of nasal antibody [160,161]. For HPIV type 1, neutralizing activity is associated with the presence of IgA [161]. A study by McIntosh and colleagues showed that in infants and children (aged 7 months to 6 years), IgA in the nasal secretions was present, but was incapable of neutralizing virus [162], suggesting that IgA is either of low avidity or directed to non-neutralizing epitopes. How T cell responses are regulated in newborns following HPIV infection has not been explored. It is clear that more research on the immune regulation that occurs in the context of infection with RVs and HPIVs is much needed given the state of the current knowledge and the impact of these infections on newborn health.

## 4. Vaccines as an Approach to Protect Infants against Respiratory Viruses

At present, there is no available vaccine for RSV, RVs or HPIVs, and none for influenza virus for infants under 6 months of age. The ability to protect newborns against these respiratory pathogens would represent a significant step forward in increasing the health of this vulnerable population. Two vaccine approaches can contribute to protection of these individuals, passive transfer of vaccine-elicited maternal antibody and direct vaccination of the newborn. During the last trimester, there is efficient transplacental transfer of antibody from the mother to the fetus that can provide protection to the newborn immediately following birth (Figure 2). However, maternal antibody will wane, leaving the infant with less protection over time. Generation of a protective response by the newborn through vaccination early following birth would allow antibody to be produced to compensate for waning maternal antibody (Figure 2). These protective mechanisms used in concert could close the window of susceptibility in infants, preventing the disease that occurs following respiratory virus infection. 

### 4.1. Maternal Vaccination

Because of the naïve status and challenges associated with the immune system of neonates, they rely heavily on maternal antibody for protection [163,164]. The maternal antibody concentration present at birth and the kinetics of decay are important factors that dictate how long these antibodies can provide protection in newborns [165,166]. The concentration of IgG transferred increases sharply during the third trimester [167,168,169] and thus, premature infants have significantly lower levels of maternal IgG than full-term infants [170]. While there is strong evidence that maternal antibody can provide protection [164], not all pregnant women have a high enough antibody response to protect infants from a given pathogen. The strategy to overcome this is to boost antibody levels by vaccination during pregnancy. At present, the World Health Organization (WHO) recommends maternal vaccination for tetanus, pertussis, and influenza [171]. It is crucial that women be vaccinated at a time that allows generation of a peak antibody response and thus optimal transfer of antibody to the infant. In a study by Englund et al., maximum transfer of antibodies to infants born to mothers who received the Haemophilus influenza type B conjugate vaccine occurred when the vaccine was administered at least one month before birth [172].

There is growing evidence of the benefit of administering the influenza vaccine to pregnant women [173,174,175,176,177]. For example, in a study in Bangladesh, vaccination of pregnant women resulted in a 63% reduction in laboratory-confirmed influenza cases in infants up to 6 months of age [173]. In another study, infants born to vaccinated mothers were 45–48% less likely to have influenza associated hospitalizations than infants born to non-vaccinated mothers [174]. In a U.S. study, the monovalent 2009 H1N1 pandemic vaccine elicited protective antibody responses in 93% of pregnant women and 87% of infants [176]. Importantly, the available data convincingly show that vaccine administration is safe in these individuals for both mother and infant [173,174,175,176].

Given the challenges of developing RSV vaccines for use in young infants, the field has focused on maternal vaccination as an approach to protect newborns. This is supported by studies showing protection against RSV correlates with the level of maternal RSV-specific antibody [143,178,179]. Neutralizing antibodies are directed to the F surface protein of RSV [180,181,182]. Recent advances in our understanding of F protein structure have paved the way for the development of new and promising vaccines [180,183,184,185,186]. Targeting F protein through vaccination is bolstered by clinical trials showing prophylactic use of the monoclonal antibodies Palivizumab and Motavizumab, which bind the F protein antigenic site II, results in reduced hospitalizations in high-risk infants [187,188]. Multiple studies in cotton rats have demonstrated that an RSV F vaccine can induce neutralizing antibodies that protect against viral challenge [183,185].

A phase 2 clinical trial using a nanoparticle vaccine containing the F protein (Novavax, Rockville, MD, USA) to vaccinate women in their third trimester was found to be safe and elicited an RSV specific antibody response in the mothers that was transferred to infants [184]. This vaccine candidate recently completed a phase 3 randomized, placebo-controlled clinical trial (NCT02624947) to evaluate protection of infants in the first months of life against clinical disease during the RSV season. Unfortunately, infants born to vaccinated mothers did not show the level of protection necessary to meet the prespecified success criterion for efficacy against RSV-associated, medically significant respiratory tract infection in infants during the first 90 days of life [189]. The results available thus far do not provide insight into the reason for the apparent failure of the vaccine to provide efficacy. Overall infants born to the vaccinated mothers had greatly increased F-specific antibody levels compared to infants born to mothers receiving the placebo (12.6-fold). However, the neutralizing activity of these antibodies was not measured, nor was the level of antibody in infants who exhibited disease compared to those who did not. In the phase 2 trial, neutralizing antibody was increased approximately 2-fold [184]. Thus, it is possible that the quality of the antibody elicited was insufficient. Maternal antibody-mediated protection of infants has been demonstrated in evaluation of infants born to non-vaccinated pregnant women with protection associated with the level of neutralizing antibody [190,191,192]. Nonetheless, in addition to neutralization, other properties of antibody can contribute to its in vivo efficacy including avidity, epitope specificity, glycosylation, and Fc-mediated functional activity. A full evaluation of the functional characteristics of antibody elicited by natural infection versus vaccination will be critical for interpreting the results of the phase 3 study and the rational design of future vaccines.

Optimizing maternal vaccination approaches will require an increased understanding of the immune response in pregnant women, ideal vaccination approaches for this population, and transplacental transfer of antibody. With regard to the first, pregnancy reflects an immune modulated state that varies across gestational age. Implantation elicits a pro-inflammatory response [193] that rapidly shifts to an anti-inflammatory state that continues through the third trimester [194]. Interestingly, anti-inflammatory markers increase progressively with gestational age, suggesting an intensifying anti-inflammatory bias through the last two trimesters of pregnancy [195]. The anti-inflammatory state present during mid-late pregnancy is associated with an expansion of Tregs [196,197] and a shift from macrophages that produce pro-inflammatory to those that produce anti-inflammatory mediators [198]. Given the immune alterations present, it is possible that distinct adjuvants or alternative routes of delivery may be selectively beneficial in pregnant women.

### 4.2. Newborn Vaccination

An optimal vaccine for newborns would generate protection following a single dose. However, this is admittedly a major challenge due to the naïve status and immune responsiveness of the infant. When the time frame required to develop a peak response followed by boosting is taken into account, the first dose of vaccine would likely need to be administered within the first few months of life to optimally protect the infant since maternal antibodies start to decline soon after birth with a half-life of 1.5–3 months [83,178,192]. The potential impact of maternal antibodies on eliciting a response in newborns following vaccination of newborns is not insignificant and will be discussed below.

#### 4.2.1. Adjuvants for Newborns

The quest for novel adjuvants remains a major focus in the field of vaccinology. Although there are many adjuvants being explored [199], this review will predominantly focus on those that target TLRs given their highly studied immunomodulatory properties [200] and promising results in newborns. TLRs represent a family of receptors found on immune cells that function by sensing PAMPs and inducing cellular signaling that leads to activation of multiple immune cell types [201].

The inactivated RSV vaccine provides an instructive model for the power of adjuvants in re-programming the immune response in newborns. Formalin-inactivated RSV was approved as a vaccine for young infants in 1969. Unfortunately, a subset of infants who received the vaccine experienced more severe disease following RSV infection [202]. We now understand the enhanced disease was the result of an aberrant response that drove eosinophilic influx [203]. The response to this vaccine has been attributed, at least in part, to poor TLR stimulation resulting in low avidity antibody that could not effectively control virus [204,205]. Provision of a TLR agonist along with the inactivated vaccine was shown to prevent enhanced disease following challenge in a mouse model [204,205]. The demonstrated ability of TLR stimulation to redirect the immune response from pathogenic to protective is particularly important in the context of newborns as it supports the ability to effectively vaccinate these individuals if the immune system is exposed to the correct stimuli.

There is mounting evidence that TLR7/8 agonists are particularly advantageous as vaccine adjuvants for infants. Philbin et al. showed the TLR8 agonist 3M-002 promotes upregulation of costimulatory signals and IL-12p70 production in neonatal cells [12]. High plasma concentrations of adenosine present in newborns have been reported to modulate TLR mediated cytokine production and cDC activation, promoting TH2 polarization and decreased TH1 responses [206,207,208]. However, neonatal monocytes stimulated with the TLR8 agonist 3M-002 were refractory to the inhibitory effects of adenosine [12]. Work from our lab also supports targeting TLR7/8. Vaccination of NHP newborns with an inactivated influenza vaccine comprised of the TLR7/8 agonist R848 conjugated to the PR8 influenza virion was a potent inducer of virus-specific IgG antibody and IFNγ-producing T cells [209,210]. TLR7/8 agonists as adjuvants have also shown promise in the context of RSV vaccines, albeit these studies were performed in adult mice. Here a nanoparticle forming thermoresponsive polymer (TRP) was used to co-deliver F protein trimers and a TLR7/8 agonist [211]. Vaccination resulted in production of high level RSV-specific neutralizing antibody that could provide protection following challenge.

CpG oligodeoxynucleotides (CpG ODNs) is a TLR9 agonist that can activate both humoral and cellular immune responses. CpG ODNs have been shown to improve immunogenicity in many viral vaccine models [199]. One example of this is co-delivery through a DNA vaccine. DNA vaccines produce antigen intracellularly to mimic the immune response that is present in a natural infection. Initial studies with these vaccines found that they exhibited reduced immunogenicity in humans and NHPs. To overcome this, a study by Ma et al. used a plasmid DNA vaccine encoding a CPG ODN motif along with the RSV F protein [212]. Vaccination of adult mice resulted in increased TH1, and antibody responses along with decreased lung pathology and viral replication following challenge [212]. The utility of CpG ODN as an adjuvant has also been reported in newborn mice. Co-administration with a vaccine against *Salmonella* or hepatitis B resulted in enhanced antibody and/or CD8^+^ T cell responses in newborn mice [213,214].

A strategy for increasing the immunogenicity of these adjuvants is the delivery of multiple TLR agonists. Simultaneous engagement of distinct TLRs has been shown to improve cDC maturation in a qualitative and quantitative fashion [21,215]. A study of human cord blood naïve CD8^+^ T cells showed increased clonal expansion and IFNγ production following stimulation with a combination of TLR2 and TLR5 agonists compared with either agonist alone [216]. The success of the tuberculosis vaccine (BCG), which is routinely delivered within 48h of birth, supports this strategy’s utility. BCG contains ligands for 5 distinct TLRs (1, 2, 4, 6, and 9) [217], and it seems likely that the ability to induce immune responses in neonates is the result of the signaling through multiple TLR.

Clearly, TLR agonists are promising adjuvants; however, there are additional strategies that may be useful to consider alongside TLR agonists. A recent study showed that a nucleoside-modified mRNA-lipid nanoparticle influenza vaccine was capable of promoting long-lived GC formation in newborn mice, resulting in improved antibody responses [218]. Virus-like particles (VLPs), an approach that utilizes empty non-infectious viral capsids that structurally mimic the conformation of virions, liposomes, and water-in-oil emulsions have also been used to improve vaccine immunogenicity [199] and may have potential in newborns. The studies evaluating the effectiveness in newborns of the adjuvants described above suggest effective adjuvants can be identified that will overcome the limitations of the newborn immune system.

#### 4.2.2. Additional Challenges Associated with Newborn Vaccination

In addition to the inherent alterations in the immune responsiveness in newborns, there are additional potential obstacles for the development of safe and effective newborn vaccines. Among these are maternal antibody inhibition, antibody-dependent enhancement, and environmental determinants that may alter the newborn’s response to vaccination.

The potential for maternal antibody to inhibit vaccine responses in young infants has been well established for measles [219,220] and pertussis [221] vaccines. Our understanding of maternal antibody inhibition with influenza vaccination is limited by the lack of administration to infants under 6 months of age; however, initial studies to evaluate immunogenicity in young infants provided indication that maternal antibody can dampen the response [222]. In this study by Halasa et al., two doses of trivalent inactivated vaccine were administered to infants aged 10–22 weeks. The presence of a high level of maternal antibody was associated with lower seroconversion rates [222]. Thus, maternal antibody will likely be a factor even when potent vaccines are identified. However, we believe this is not insurmountable if we can understand at a mechanistic level what is responsible for the inhibition.

The extent to which maternal antibody impacts the response can vary among vaccines as well as individuals. This is undoubtedly a contributor to the existing uncertainty around the importance of the various mechanisms that have been associated with maternal antibody inhibition. It seems likely that factors such as differences in the quantity, characteristics, and/or quality of maternal antibody present at the time of vaccination affect the extent of inhibition. Multiple mechanisms have been proposed for this effect including epitope masking, binding of antibody to the inhibitory FcγRIIB on B cells, and antigen removal by macrophages [223]. A recent study by Siegrist and colleagues shed light on a mechanism that may be at play during influenza vaccination. They found evidence that maternal antibodies prevent the differentiation of GC B cells into plasma cells and memory B cells [224], providing an opportunity to selectively target impediments at this step. Further study is required to identify the most important factors limiting newborn responses for individual vaccines and approaches, e.g. adjuvants or route of delivery, that may overcome them. While these inhibitory effects are important potential regulators of vaccine responses, it is worth noting that there are circumstances under which maternal antibody can also promote immune responses [223,225,226,227]. Thus, maternal antibody regulation of immunity in newborns is complex and an area deserving of additional study.

While antibodies play a critical role in limiting infection through neutralization, non-neutralizing antibodies are also generated. Although these can be beneficial, there are circumstances under which they promote virus infection and disease, a phenomenon known as antibody-dependent enhancement (ADE). In ADE, non-neutralizing low avidity antibodies bind to virus particles allowing for Fc receptor mediated entry of viruses into macrophages and monocytes, which can alter virus tropism and increase viral pathology [228]. ADE has been routinely documented in dengue virus infection [229]; however, ADE associated with increased disease severity has also been reported for influenza virus and RSV [230,231,232,233] as well as other viruses [229,230,231,234,235,236,237,238]. In pigs vaccinated with inactivated H1N1 and/or H1N2 vaccines, there was demonstrated protection against homologous viral challenge, while heterologous viral challenge resulted in disease enhancement [239,240]. There is also a report of maternally derived antibodies causing increased disease in piglets following heterologous influenza virus challenge [241]. Interestingly, in humans vaccinated with the 2008-09 trivalent inactivated influenza vaccine, there were cases of enhanced illness following infection with the pandemic H1N1 strain compared to those who did not receive the vaccine [242,243]. The mechanism responsible for the enhanced disease is unclear and likely multi-factorial, but it is intriguing to speculate that there may be a role for ADE. These studies highlight the importance of evaluating the potential for generation of antibodies that may promote enhanced disease when developing vaccines for infants.

The newborn response to vaccination can vary across geographic locations. An example is found in the BCG vaccine that is administered to newborns. Vaccine-induced responses in infants from Malawi, the Gambia, and the UK where compared at 3 months following vaccination [244]. This study revealed a hierarchical response with infants from the UK exhibiting the highest cytokine response to the vaccine followed by infants from the Gambia and finally Malawi [244]. These results are intriguing and while many differences are present in these populations, one possibility is that differences in the gut microbiota contribute to vaccine responsiveness. It is now known that environmental factors and host genetics can influence the establishment of the microbiome and this in turn drives immune system development [245] and the response to vaccination [245,246,247]. The gut microbiota appears to be a natural source of adjuvants that are important in priming neonatal immune responses [245]. A study using germ-free and antibiotic-treated mice that were vaccinated with ovalbumin together with complete Freund’s adjuvant revealed reduced serum antibody titers in germ-free compared to conventionally raised mice [248]. This response was restored when the germ-free mice were colonized with the microbiota of conventionally raised mice. Similar impairments in immunogenicity were observed in germ-free mice that were vaccinated with inactivated influenza vaccine [249].

Huda et al. investigated whether the gut microbiota composition in infants could predict vaccine responses. Stool samples were taken from 46 Bangladeshi infants at 6, 11 and 15 weeks of age and the composition of the microbiota was analyzed. The vaccine-specific T cell response as well as antigen-specific IgG was measured. Their data showed that the presence of *Actinobacteria* was associated with improved vaccine responses [250]. Further, greater microbial diversity in infants was associated with neutrophilia and lower vaccine responses [250]. This study shows that differences within the microbiome may affect the immune response to vaccination. The composition of the microbiome is directed by a myriad of factors including gestational age, mode of delivery, diet, hygiene, antibiotics, environment, and probiotics. The results described above are unquestionably intriguing and more work is needed to fully understand the interplay between the microbiome, immune development, and vaccine responses in newborns. If future studies support the ability of an optimal microbiome to generally support vaccine responses in infants, active modulation in early infancy may prove a useful approach to increasing vaccine effectiveness.

## 5. Concluding Remarks

Newborn vulnerability to severe disease following respiratory virus infection is a significant public health concern. In the case of the most clinically relevant pathogens (influenza virus, RSV, HPIVs, RVs), vaccines are not yet available for newborns. Thus, at present there is a focus on maternal vaccination to protect infants against influenza virus and RSV. Vaccines for these two pathogens for adults are either approved or in clinical trials. While maternal antibody provides benefit, waning levels will leave the infant increasingly unprotected in the months following birth. Thus, complementing maternal vaccination with early delivery of a vaccine directly to the infant is an appealing approach. Early vaccination would allow for administration of a prime and boost dose of the vaccine that will likely be required to achieve protective antibody levels in the infant. While a number of experimental adjuvants show promise for newborns, a more in depth understanding of the immune system of these individuals will be critical to the development of optimal adjuvants and vaccine delivery approaches. This will be facilitated by use of animal models that most closely reflect the human newborn immune system, e.g. NHP. Use of these models will be complemented by direct analysis of human newborns. The advances in experimental approaches allow increasingly more information to be garnered with the limited sample that can be obtained from these individuals. Although the challenges are significant, they are not insurmountable, and we are continually nearing the goal of developing vaccine approaches that can protect young infants from infection and disease.

## Figures and Tables

**Figure 1 vaccines-08-00558-f001:**
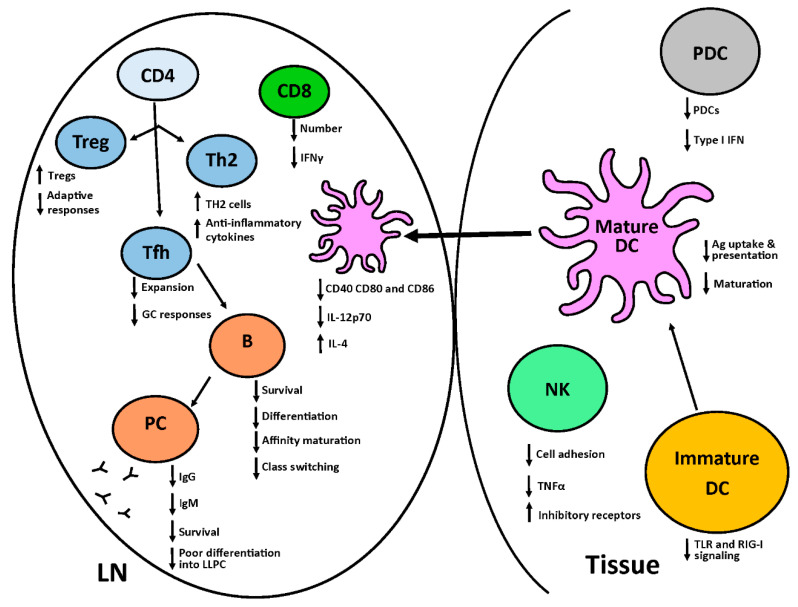
The neonate immune system has alterations that span both the innate and adaptive arms, both of which are important in viral clearance. Within the innate arm, NK cells have functional deficiencies that include decreased cell adhesion due to a reduction in selectins, reduced TNFα production, and increased inhibitory receptors. The number of neonatal PDCs is reduced as well as the amount of type I IFN produced from these cells. Potent adaptive immune responses are dependent on the capacity of DCs to undergo maturation together with the robust activation, differentiation and survival of T and B cells. DCs from newborns are challenged in this process, as they have decreased capacity to respond to PAMPs through TLR and RIG-I receptors resulting in impaired costimulatory molecules (CD40, CD80 and CD86) expression as well as diminished IL-12p70 production that is essential for the differentiation of CD4 TH1 cells and CD8^+^ T cells. There is also increased IL-4 production from newborn DCs. The low levels of IL-12p70 accompanied by increased IL-4 production skews the differentiation of CD4^+^ T cell towards TH2 cells. Decreased IL-12p70 impacts the CD8^+^ T cell differentiation leading to decreased production of IFNγ. Neonates also have heightened Treg responses. There are functional defects in neonatal Tfh cells including limited expansion leading to poor GC responses and reduced production of high affinity, isotype switched Ab.

**Figure 2 vaccines-08-00558-f002:**
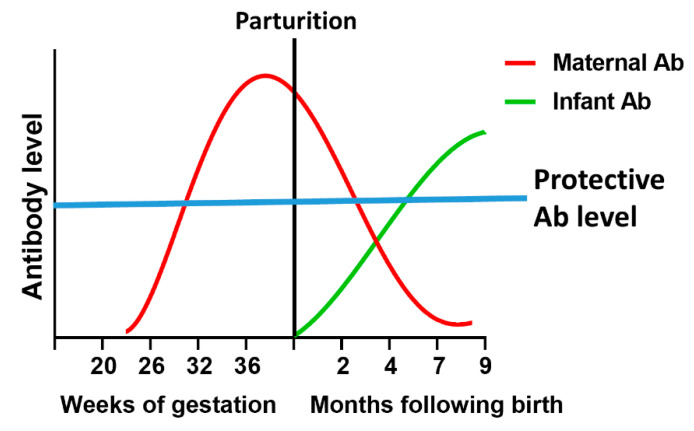
Eliminating the window of vulnerability of neonates to respiratory infection. The window of vulnerability describes the timeframe wherein maternal-derived antibody and infant-derived antibody levels are low and as a result there is minimal protection against pathogen infection. The combination of maternal antibody, which provides protection early following birth and infant-derived antibody, generated by vaccination in the first months following delivery, would limit the window of vulnerability. Maternal antibody level can be enhanced by vaccination during pregnancy at a time that allows maximal production and transplacental transfer of high affinity IgG (red line). Vaccination and boost of newborns soon after birth allows the production of antibody by the infant to reach protective levels (green line) prior to waning of maternal antibody to a level that is no longer protective (blue line).

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
