# Peer review of "Challenges for the Newborn Immune Response to Respiratory Virus Infection and Vaccination"

_vaccines, 2020, doi:10.3390/vaccines8040558_

Round 1

Reviewer 1 Report

In the manuscript “Challenges for the newborn immune response to respiratory virus infection and vaccination”, the authors aim at reviewing the differences in the neonatal immune system, the newborn response to respiratory virus infection, and the modalities of using vaccines to protect infants against respiratory viruses. The manuscript is extremely interesting, in particular the part focusing on vaccine potential and strategies.

The manuscript is well written, but some sentences tend to be relatively long and complicated, thus difficult to understand. Given the topic, the review will likely attract a wide audience and the authors should keep that in mind, as all readers will not be specialized immunologists or virologists.

Specific points:

  • Be careful with acronyms. Some are defined after they first appeared (DC), others are not defined (APC).
  • Lines 28-29 “The increased disease severity to these viruses decreases with age…”: this sentence is awkward. Having ‘increased something’ that ‘decreases’ is really not nice.
  • Line 55: ‘TH2 bias’ is not meaningful. The rest of the sentence is clearly stating the effects (‘reduced T cell activation, impaired B cell differentiation and survival, decreased quantity and quality of antibody), but bias does not clearly express what happens. Perhaps is it also due to the fact that the differences betweenTH2/TH1 response are not clearly explained in the manuscript?
  • Lines 81-84: the link between impaired NK cell function and reduction in selectins or decrease in TNF-alpha is not clear here. It becomes clear when one looks at Figure 1. But figures are never referred to in the text.
  • Figure 1 legend: APRIL is not on the figure so it should not be in the legend.
  • Part 2 in general: the authors describe the differences between newborns/infants and adults. Sometimes, when ready the manuscript, it would be useful to know when each aspect of the immune response reaches a level which is comparable to adults or functional (at which age, under which circumstances).
  • Lines 124-125: the examples of lck, ZAP-70 and AP-1 are not clear. Decrease levels or reduction, ok, but then… what? What does it mean?
  • Part 3: in general in this part, the authors should perhaps present the clinical aspects first to help the reader follow. What is the clinical presentation in newborns/infants compared to children with a mature immune system or compared to adults (but which adults? Without co-morbidities?)? What is a severe form? Then, it would be nice to know what is the ‘normal’ immune response, i.e. not in newborns/infants. Finally why the differences in the newborn immune system might explain why the clinical presentation is more severe.
  • Line 177 “it is not surprising that newborns struggle to control influenza virus infection”: this sentence for example is not clear. What does it mean? Can the authors draw a parallel with older adults with severe forms in the immune response profiles?
  • Lines 198-199 “A study by You et al infected 7-day old…finding a decrease…”: this sentence is clumsy. The study did certainly not infect mice. The authors of the study did.
  • Line 243: what do the authors mean by “the representation of antibodies to the individual sites”? not clear.
  • Line 246: “converged over time”? not clear.
  • Line 258: check taxonomy of RSV. Not in the Paramyxoviridae family anymore.
  • Lies 266-267: “This analysis revealed a dysregulated immune profile with robust increases in expression of genes associated with…” Not clear.
  • Lines 315-317: it is surprising to see RV used in a singular form given the large variety of types that exist.
  • Lines 400-402: not clear
  • Line 404-405: sentence is weird. “the goal… would be THE delivery of…”?
  • Lines 449-450 “A study by Ma et al adjuvanted a plasmid …”: this sentence is clumsy. The study did not adjuvant anything.
  • Lines 468-469: this is not clear why.
  • Line 531 “greater microbial diversity in infants leads to increased inflammation and lower vaccine response”: this is an interesting point but it is not clear how the authors link this sentence to what was presented before. How was the microbial diversity evaluated? Is this conclusion only concerning study 225 or is it also a possible explanation for the difference observed between UK and Malawi? And what are the consequences in terms of vaccine development for newborns? Does it mean that vaccines formulation/adjuvant composition should be adapted depending on the intended population?
  • Lines 549-552: very long sentence, without clear insightful information.

Author Response

We thank the reviewers for their careful reading of our manuscript and the suggestions offered for improvement.  These changes have significantly increased readability and completeness of our review.

Reviewer #1

In the manuscript “Challenges for the newborn immune response to respiratory virus infection and vaccination”, the authors aim at reviewing the differences in the neonatal immune system, the newborn response to respiratory virus infection, and the modalities of using vaccines to protect infants against respiratory viruses. The manuscript is extremely interesting, in particular the part focusing on vaccine potential and strategies.

The manuscript is well written, but some sentences tend to be relatively long and complicated, thus difficult to understand. Given the topic, the review will likely attract a wide audience and the authors should keep that in mind, as all readers will not be specialized immunologists or virologists. We appreciate this reminder that it is easy to become so comfortable with our field-specific  vocabulary that it can result in a review that is less accessible for others. We have worked to make the review more reader friendly and to increase readability.

Specific points:

  • Be careful with acronyms. Some are defined after they first appeared (DC), others are not defined (APC). We thank the reviewer for pointing this out. We have checked carefully to ensure that all acronyms have been defined at first use.
  • Lines 28-29 “The increased disease severity to these viruses decreases with age…”: this sentence is awkward. Having ‘increased something’ that ‘decreases’ is really not nice. We thank the reviewer for this suggestion. This sentence has been edited for readability. Lines 28-30.

  • Line 55: ‘TH2 bias’ is not meaningful. The rest of the sentence is clearly stating the effects (‘reduced T cell activation, impaired B cell differentiation and survival, decreased quantity and quality of antibody), but bias does not clearly express what happens. Perhaps is it also due to the fact that the differences betweenTH2/TH1 response are not clearly explained in the manuscript? We appreciate the reviewer’s comment about TH2 bias. We have now chosen to keep the summary statement around T cells here more general (line 56). The specifics regarding Th1/Th2 differentiation are discussed below.

  • Lines 81-84: the link between impaired NK cell function and reduction in selectins or decrease in TNF-alpha is not clear here. It becomes clear when one looks at Figure 1. But figures are never referred to in the text. We appreciate the reviewer’s suggestion and have edited this section (starting line 75) to provide more clarity around function. We apologize for the omission of the Fig 1 citation. This has been added.

  • Figure 1 legend: APRIL is not on the figure so it should not be in the legend. This has been removed from the figure legend.
  • Part 2 in general: the authors describe the differences between newborns/Infants and adults. Sometimes, when ready the manuscript, it would be useful to know when each aspect of the immune response reaches a level which is comparable to adults or functional (at which age, under which circumstances). Maturation of the immune system to one which reflects adults is a challenging issue as there is some variation across reports. However, where available, we have added this information throughout.

  • Lines 124-125: the examples of lck, ZAP-70 and AP-1 are not clear. Decrease levels or reduction, ok, but then… what? What does it mean? Lck, ZAP-70, and AP-1 are important TCR signaling molecules or transcription factor important for the activation of T cells. We have expanded this section to provide additional insight to facilitate understanding of these changes. Lines 142-151

  • Part 3: in general in this part, the authors should perhaps present the clinical aspects first to help the reader follow. What is the clinical presentation in newborns/infants compared to children with a mature immune system or compared to adults (but which adults? Without co-morbidities?)? What is a severe form? Then, it would be nice to know what is the ‘normal’ immune response, i.e. not in newborns/infants. Finally why the differences in the newborn immune system might explain why the clinical presentation is more severe. We have carefully considered the comments of the review. We have expanded the discussion of the clinical aspects of the disease that occurs in newborns for each of the viruses discussed in the review, including the severe manifestations. This has been placed at the beginning of each section as suggested. With regard to the alterations that contribute to increased disease in newborns, it is likely the combination of the multiple impairments that are present.

  • Line 177 “it is not surprising that newborns struggle to control influenza virus infection”: this sentence for example is not clear. What does it mean? We appreciate this concern, however, this sentence was removed during the editing of this section.

  • Lines 198-199 “A study by You et al infected 7-day old…finding a decrease…”: this sentence is clumsy. The study did certainly not infect mice. The authors of the study did. This has been edited in the manuscript. Lines 224-25

  • Line 243: what do the authors mean by “the representation of antibodies to the individual sites”? not clear. We appreciate that the terminology here was not precise. We have significantly modified this section. Lines 269-283

  • Line 246: “converged over time”? not clear. Our studies in the NHP model revealed altered immunodominance patterns in the early IgM anti-HA response in newborns compared to adults. However, over time the patterns became similar between the neonate and the adult. We have edited the manuscript for more clarity. Lines 269-283

  • Line 258: check taxonomy of RSV. Not in the Paramyxoviridae family anymore. We appreciate this correction; however, based on the suggestions of reviewer 2 we have removed this detail.

  • Lines 266-267: “This analysis revealed a dysregulated immune profile with robust increases in expression of genes associated with…” Not clear. We have altered this sentence to “This analysis revealed a dysregulated immune profile induced by RSV infection, showing overexpression of genes associated with neutrophil activation, type I IFN and systemic inflammatory markers when compared to healthy age-matched infants [132]. Further, infected infants exhibited reduced expression of genes regulating T and B cell activation [132].”

  • Lines 315-317: it is surprising to see RV used in a singular form given the large variety of types that exist.  We agree that the plural form is more appropriate and have changed this throughout.

  • Lines 400-402: not clear. We apologize that we did not define the terms M1 and M2. These are alternative states of macrophages differentiation. We have replaced this terminology with a description of the cells. Lines 452-453

  • Line 404-405: sentence is weird. “the goal… would be THE delivery of…”? We have made this change in the manuscript. Line 457

  • Lines 449-450 “A study by Ma et al adjuvanted a plasmid …”: this sentence is clumsy. The study did not adjuvant anything. We have reworked the sentence to increase clarity. Lines 500-503

  • Lines 468-469: this is not clear why. We appreciate the reviewer’s comment and have addressed this in the manuscript “The studies evaluating the effectiveness of the vaccine delivery approaches and adjuvants described above suggest effective adjuvants can be identified that will overcome the limitations of the newborn immune system.”

  • Line 531 “greater microbial diversity in infants leads to increased inflammation and lower vaccine response”: this is an interesting point but it is not clear how the authors link this sentence to what was presented before. How was the microbial diversity evaluated? Is this conclusion only concerning study 225 or is it also a possible explanation for the difference observed between UK and Malawi? And what are the consequences in terms of vaccine development for newborns? Does it mean that vaccines formulation/adjuvant composition should be adapted depending on the intended population? We have reworked this paragraph to provide additional information and insights. In the Bangladesh study, microbial diversity was evaluated by amplification and sequencing of bacterial 16S ribosomal RNA. The conclusion from this study is that changes in the microbiome may alter the immune response to vaccination. It will be important to understand the role of the microbiome within the neonate immune response as well as what bacteria are involved in generating effective responses to vaccination in newborns. The implications for this are discussed in the manuscript. Paragraph beginning line 518

  • Lines 549-552: very long sentence, without clear insightful information. We appreciate this feedback and have shortened this sentence to increase clarity. Lines 613-616

Reviewer 2 Report

In this manuscript, the authors should focus on providing a review of the characteristics of the newborn immune response to respiratory virus infection and the challenges for neonatal vaccination. It is recommended to remove irrelevant and tedious descriptions, e.g. depicts for influenza in Lines 172-175 and depicts for RSV in Lines 256-258.

  1. Human dendritic cells mainly consist of two developmentally-distinct lineages, conventional (cDCs) and plasmacytoid DCs (pDCs). Please first describe the features of neonatal cDCs in Section 2.1 and simply depict the impaired functional capabilities of DCs in Section 2.2.
  2. In Lines 76-79, it is recommended that the author do not have to describe how NK cells exert the function of removing pathogens. The author should pay more attention to summarize the variation of NK numbers and which functions of neonatal NKs are impaired.
  3. Monocytes and macrophages also play a key role in pathogen recognition and eradication throughitheir phagocytic, antigen-presenting and cytokine-secreting abilities. Neutrophils provide the first line of defense against pathogens through phagocytosis, release of toxic substances and generation of neutrophil extracellular traps (NETs). Please describe the differences of monocytes, macrophages and neutrophils in neonatal innate immune system, compared to adult.
  4. In Line 236, what is BALT? Please give the full name of BALT.
  5. In fact, Novavax had already announced that ResVaxTM, RSV F vaccine for infants via maternal immunization, failed to meet the trial’s primary efficacy endpoint in 2019, which is a heavy blow to RSV vaccine research. Please the authors share your constructive opinions for maternal vaccination in the light of the failure of Novartis RSV vaccine.

Author Response

We thank the reviewers for their careful reading of our manuscript and the suggestions offered for improvement.  These changes have significantly increased readability and completeness of our review.

Reviewer #2

In this manuscript, the authors should focus on providing a review of the characteristics of the newborn immune response to respiratory virus infection and the challenges for neonatal vaccination.

It is recommended to remove irrelevant and tedious descriptions, e.g. depicts for influenza in Lines 172-175 and depicts for RSV in Lines 256-258. We have removed this information as requested as it is not critical. We have worked to keep this as succinct as possible and have considered throughout if there is opportunity to be more concise.

  1. Human dendritic cells mainly consist of two developmentally-distinct lineages, conventional (cDCs) and plasmacytoid DCs (pDCs). Please first describe the features of neonatal cDCs in Section 2.1 and simply depict the impaired functional capabilities of DCs in Section 2.2. Thank you for this suggestion, we have moved the features of cDC to 2.1 and only included the impaired functional capabilities of DCs in section 2.2.

  1. In Lines 76-79, it is recommended that the author do not have to describe how NK cells exert the function of removing pathogens. The author should pay more attention to summarize the variation of NK numbers and which functions of neonatal NKs are impaired. As suggested by the reviewer, we have limited our discussion of NK function and focused on the differences in adults and newborns. Lines 92-104

  1. Monocytes and macrophages also play a key role in pathogen recognition and eradication through their phagocytic, antigen-presenting and cytokine-secreting abilities. Neutrophils provide the first line of defense against pathogens through phagocytosis, release of toxic substances and generation of neutrophil extracellular traps (NETs). Please describe the differences of monocytes, macrophages and neutrophils in neonatal innate immune system, compared to adult. We agree that these areas did not receive adequate attention in the original submission. We have now expanded these sections. Lines 61-76

  1. In Line 236, what is BALT? Please give the full name of BALT. The definition and description of this important lymphoid tissue has been added to the manuscript. Lines 261-266

  1. In fact, Novavax had already announced that ResVaxTM, RSV F vaccine for infants via maternal immunization, failed to meet the trial’s primary efficacy endpoint in 2019, which is a heavy blow to RSV vaccine research. Please the authors share your constructive opinions for maternal vaccination in the light of the failure of Novartis RSV vaccine. We appreciate the reviewer pointing out this important result. We have now included a discussion of the trial result and the implications for further pursuit of this goal. Lines 423-443